materials science

POM, ageing, infrared spectroscopy (IR), heat treatment, grain size

**Authors for correspondence:**
Hai-Ying Ni
e-mail: hyni@163.com
Jun Chen
e-mail: cschen@vip.163.com

# Influence of early thermal-oxidative ageing on the structure and properties of polyoxymethylene copolymer

Yuan-Jin Pang[1], Wen-Shuai Xu[1], Ben-Tengzi Yang[2], Hai-Ying Ni[2] and Jun Chen[2]

[1]ShanDong Shinva Saraya Biotechnology Co., Ltd, People's Republic of China
[2]Department of Polymer Processing and Engineering, College of Polymer Science and Engineering, Sichuan University, Chengdu, Sichuan 610065, People's Republic of China

Y-JP, 0000-0003-3675-0834; JC, 0000-0002-1733-8863

Thermal-oxidative ageing of polyoxymethylene (POM) copolymer in the oven at 100°C for 1, 2, 3, 5, 7, 10, 14 and 21 days and the influence of early thermal-oxidative ageing on POM structure and properties were studied by means of wide-angle X-ray diffraction, Fourier transform infrared spectroscopy, differential scanning calorimetry and tensile test. Based on the results, we found that the early thermal-oxidative ageing of POM copolymer can be divided into three regions. The region I is the initial 3 days. In this region, some molecular chains rearranged, resulting in internal stress relaxation, increase of crystallinity degree and grain size due to the perfection of crystal structure; both extended chain crystal (ECC) and folded chain crystal (FCC) increased and ECC grew faster than FCC. The region II is from 3 days to 10 days, and in this region, chain scission took place in amorphous region and led to chemi-crystallization. The region III is after 10 days. In this region, the structure and performance of POM copolymer reached a stable situation at this stage. In this work, the difference between skin and core were also analysed.

## 1. Introduction

Polyoxymethylene (POM) is a well-known engineering thermoplastic with outstanding tribological characteristic, low creep deformation as well as high specific strength close to metals [1,2] and has been widely employed in the automobile industry. However, the particular molecular structure of POM leads to a fatal weakness, i.e. terrible thermal stability, which limits the application of POM greatly. Obviously, investigations

This article has been edited by the Royal Society of Chemistry, including the commissioning, peer review process and editorial aspects up to the point of acceptance.

on the thermal-oxidative ageing behaviours of POM are of great significance in industries and in scientific research.

The introduction of some thermally stable monomers (ethylene oxide, dioxolane etc.) into the chains of POM can improve its thermostability because the driving processes slow down [3] and the depression of the degradation sensitivity of POM copolymers has been investigated by several authors [4–6]. Now, the ageing procedure of POM copolymer is generally described as follows. Under thermal-oxidative loading, the comonomer units are damaged at first [7,8], then the random scission of the main chain (repeating carbon-oxygen linkage) takes place [9]. If the ageing temperature is very high or the ageing time is very long, formaldehyde and formic acid will emerged, which play a catalytic role in the depolymerization of POM and can accelerate the degradation dramatically [10]. There is no latest research progress on the early thermal-oxidative ageing of POM. The current research point is mainly to improve the ageing conditions, such as adding hydrogen peroxide solution and performing ultraviolet irradiation to accelerate the performance change of POM [11,12].

Previous investigations have revealed that during the ageing of POM at the temperatures below its melting points (usually between 80°C and 1400°C), the properties generally show a relatively significant change before 500 h and then keep stable even after a long period of ageing [13,14]. Ageing in a natural environment also shows a similar trend, that is, damage generally occurs in the first 2 years and then the material properties keep stable until 10 years [15], which is in the agreement with the Williams–Landel–Ferry (WLF) expression about the mechanical relaxation phenomenon of polymer materials. So in this work, we focus on the early thermal-oxidative ageing of POM copolymer, aiming at describing the relationship between structure and properties in this period. As the environmental temperature of automobile indoor products is below 100°C, our ageing experiment was conducted at 100°C and the longest ageing time was 21 days (about 500 h). Wide-angle X-ray diffraction (WAXD), Fourier transform infrared spectroscopy (FTIR), differential scanning calorimetry (DSC) and mechanical tests were employed to characterize the changes in structure and properties of POM copolymer during ageing. It was found that the early thermal-oxidative ageing of POM copolymer can be divided into three regions. The difference between the core layer and skin layer was also considered.

# 2. Experimental

## 2.1. Materials and sample preparation

Commercial POM copolymer, M90-44 (Polyplastics company, Japan), containing 5% polytetrafluoroethylene ($D_{50} = 1.5\,\mu m$) powder and 3% silicone oil was used. After drying at 110°C for 3 h to prevent the degradation before processing, samples were injection moulded into specimens with a dimension of $150 \times 10 \times 4$ mm (length × width × thickness) using an injection moulding machine (PS40E5ASE, Nissei, Japan) with the injection temperature of 200°C, the mould temperature of 80°C and a moulding cycle of 50 s.

## 2.2. Thermal-oxidative procedures

The ageing procedure was carried out according to ASTM D 5510-94, 2001. As the thickness of specimens is greater than 0.25 mm, the oven (Shanghai Laboratory Instrument Works Co., Ltd, China) was forced ventilation. Specimens were put into the oven with the temperature set to 100°C. After 1, 2, 3, 5, 7, 10, 14 and 21 days of ageing, the specimens were taken out to perform mechanical tests, FTIR, DSC and WAXD characterization. The specimens without ageing were also prepared and characterized for comparison.

## 2.3. Characterizations

The tensile strength and elongation at break of the specimens were tested on a universal testing machine (Shimadzu AGS-J) in the tensile mode at a crosshead speed of 50 mm min$^{-1}$ at room temperature. At least five specimens were tested and the average values were reported. The impact strength is tested according to GB/T 1843, and the notch depth is 2 mm; the compressive strength and bending strength are tested according to GB/T 9341.

WAXD profiles were recorded on a Philips X' pert Pro MPD diffractometer (The Netherlands) using a CoK$\alpha$ radiation source ($\lambda = 0.179$ nm, 35 kV, 30 mA) in the scanning angle range of $2\theta = 5$–50° at a scan speed of 3° min$^{-1}$.

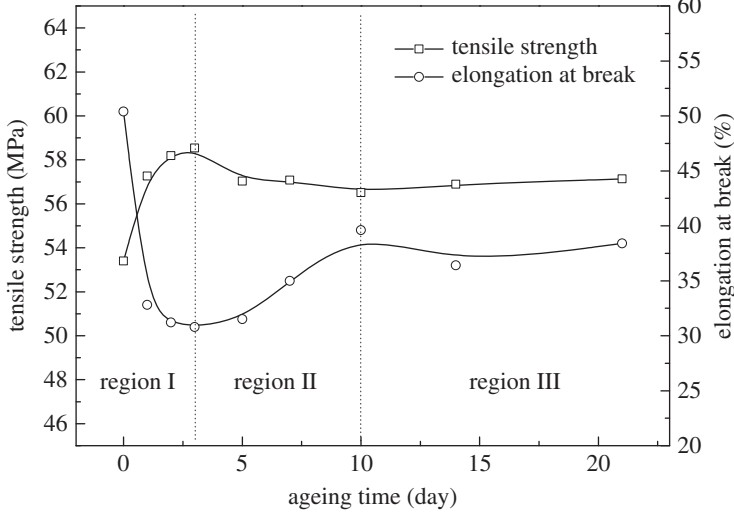

**Figure 1.** Tensile strength and elongation at break of POM after different ageing time at 100°C.

In order to study the growth of carbonyl bands (1735 cm$^{-1}$) and other bands which were assigned to extended chain crystals (ECCs) and folded chain crystals (FCCs), Fourier transform infrared (FTIR) spectroscopy was performed on a Nicolet 6700 FTIR spectrometer (Nicolet Instrument Company, USA) with the resolution of 4 cm$^{-1}$ between 650 and 4000 cm$^{-1}$ in the mode of attenuated total reflection. During data processing, the linear baseline corrections were performed in the region of 1260–550 cm$^{-1}$.

Thermal analysis was carried out on a differential scanning calorimeter (DSC) with a DSC Q200 (TA Instruments, USA) under a nitrogen gas flow of 50 ml min$^{-1}$. To reveal the difference between the surface layer and the core layer of the specimens, samples of approximately 5 mg cut at the surface and in the core were characterized at a heating rate of 10°C min$^{-1}$ from 40 to 300°C.

SEM was used to observe the fracture of the sample with JSM–5900LV (Japan Corporation, Japan), immerse the injection-moulded sample in liquid nitrogen for 20 min, after quenching, then etching with N, N-dimethylacetamide. Under the condition of the emission voltage of 5 kV, the SEM was used to observe the cross-sectional morphology of the sample and take pictures.

# 3. Results and discussion

## 3.1. Mechanical properties and region division

Macroproperties can reflect the change of structures of materials intuitively and the mechanical properties were commonly used because of its simple operation and good repeatability, so the mechanical properties of POM copolymer after ageing for 1, 2, 3, 5, 7, 10, 14 and 21 days at 100°C were tested first.

The tensile strength and elongation at break of POM copolymer after ageing are shown in figure 1. The tensile strength goes upward obviously during ageing from 0 to 3 days, then begins to decline until 10 days and after that the curve becomes flat. The elongation at break shows a contrary tendency. From 0 to 3 days it decreases sharply, and then increases to a certain degree until 10 days and after that it finally stabilizes.

According to the tendency shown in figures 1 and 2, the early thermal-oxidative ageing process of POM copolymer can be divided into three regions. The region I is from 0 to 3 days. Clearly, the tensile strength increases and elongation at break decreases in this region. The region II is from 3 to 10 days, and the tensile strength decreases and elongation at break increases respectively. The region III is from 10 to 21 days. In this region, the change of mechanical properties of POM is inconspicuous. The changes are caused by the structure of POM; it will be discussed in detail in the following sections.

## 3.2. WAXD analysis

It has been widely reported that POM crystal has a hexagonal unit cell with the unit cell dimensions of $a = b = 4.45$ Å and $c = 17.3$ Å [16]. The molecular chains are arranged in a 9/5 helix where $a$ and $b$ axes are on the same plane and the chains are aligned parallel to the $c$ axis of the crystal [17].

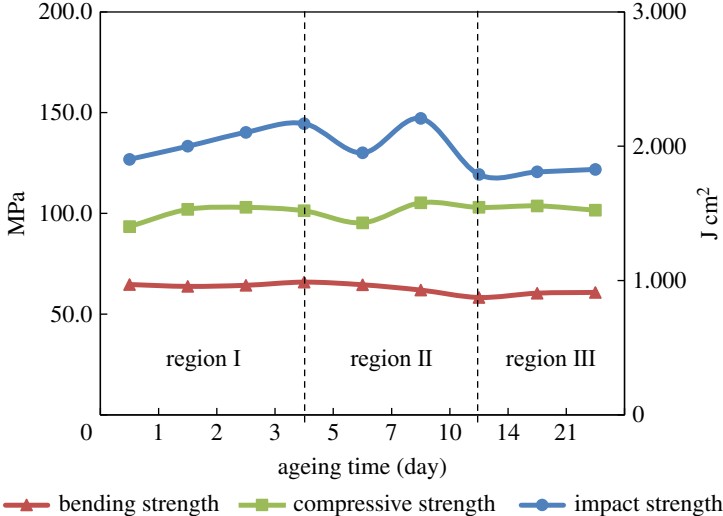

**Figure 2.** Bending strength, compressive strength and impact strength of POM after different ageing time at 100℃.

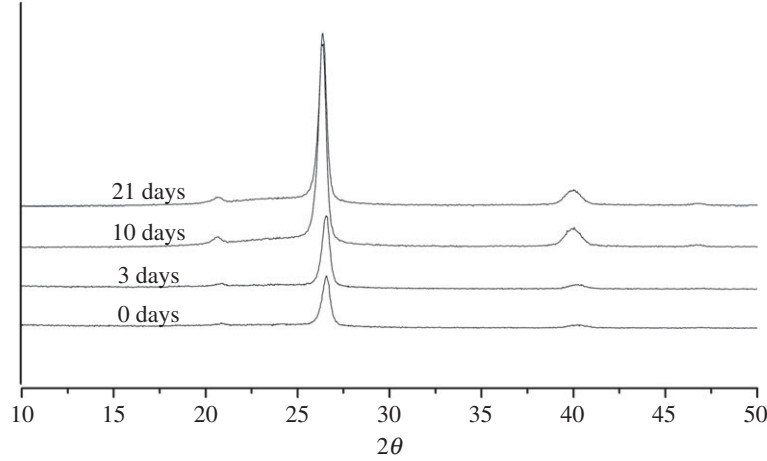

**Figure 3.** WAXD diffraction patterns of POM copolymer after ageing for various time.

Figure 3 gives the WAXD diffraction patterns of POM copolymer after ageing for various time. The diffraction peaks in the $2\theta$ scan are 26.4, 40.2 from the (100), (105) crystal planes of POM crystals, respectively [18]. The amorphous diffraction is at about 220° ($2\theta$). It can be seen that during ageing, the positions of the characteristic diffraction of POM crystals remain unchanged, indicating that the thermo-oxidative ageing does not change the hexagonal crystal structure [19]. But clearly, the diffraction intensity of these characteristic diffraction peaks changes greatly.

According to previous studies, POM undergoes heat treatment at the beginning of ageing [20,21]. In this process, the molecular chains rearrange, leading to relaxed internal stress developed in the rapid cooling process of injection moulding, increased crystallinity degree and perfection of crystal structure [13].

From the WAXD patterns, the crystallinity $X_w$ can be calculated, and the calculated data are shown in figure 4a. Based on the width of the diffraction peaks, the crystal grain size can also be calculated by Scherrer equation

$$L_{h\,k\,l} = \frac{k\lambda}{\beta\cos\theta},$$

where $L_{h\,k\,l}$ is the grain size perpendicular to the crystal plane (100), $k$ is a constant, $\lambda$ is the wavelength of X-ray, $\theta$ is the diffraction angle and $\beta$ is the width of diffraction peak [22]. The calculated data are given in figure 4b. It can be seen that both the crystallinity and crystal grain size increase from 0 to 3 days, in good agreement with the change of tensile strength in this period. It has been reported that the crystal grain size increases during heat treatment [23]. Generally, heat treatment can improve the performance of materials and release internal stress.

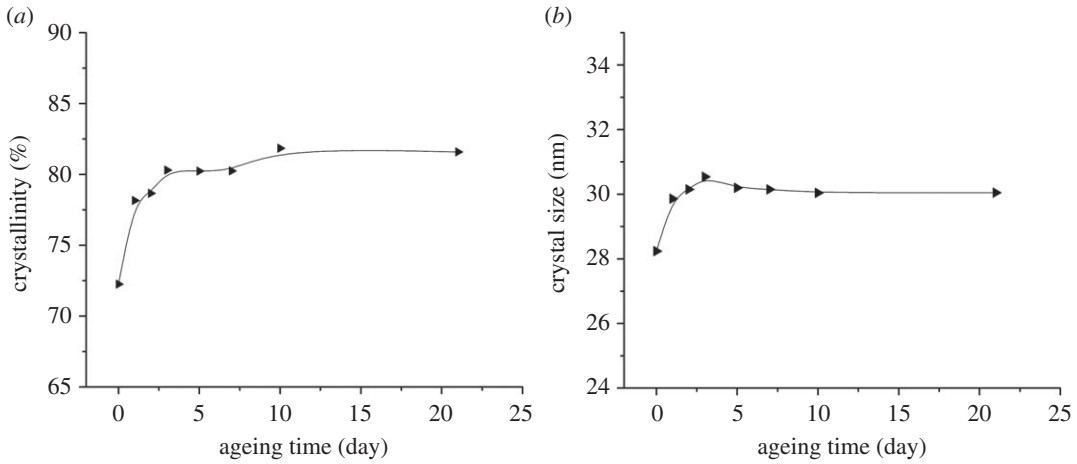

**Figure 4.** The change of crystallinity $X_w$ (a) and crystal size (b) during ageing.

Mechanical properties depend on chemical structure, molecular weight and distribution, branched and cross-linked, stress concentration, crystallization and orientation, and so on. The first three factors were not changed or happened in this period obviously and stress concentration was released. Figure 4a also pointed out that crystallinity was increased, so the decrease of tensile strength must be induced by the decrease of molecular weight. We can easily prove that chain scission happened in region II, and it happened just after the ageing of samples for 3 days [24]. The chain scission makes for rearrangement of molecular chain and leads to chemi-crystallization of POM copolymers. The chemi-crystallization process of POM only occurred in the amorphous region [3] and has been described in numerous studies on the degradation of crystalline polymers [25,26].

In region II, the elongation at break increases, because the decrease of crystal size (figure 4b) can improve the ductility of the injection-moulded samples [27,28]. The increase of ductility indicates that the embrittlement does not occur in this region. As reported, the ageing of POM will suffer from oxidation first, then chain scission and embrittlement [14]. So here our results demonstrate that the early thermal ageing does not embrittle the POM copolymer.

In region III, figure 1 shows that there were no changes in tensile strength or elongation at break, and figure 4 also shows that crystallinity and crystal grain size stay stable, suggesting that POM copolymer has reached a stable situation after 10 days of ageing.

## 3.3. FTIR analysis

### 3.3.1. Change of FCC and ECC

It has been revealed that the short-time ageing in our work increases the crystallinity of POM copolymer. It is well known that ECC and FCC are the typical crystalline morphologies of POM [29]. FCC of POM can be observed from the dilute solution [30], whereas ECC is generally found in the cationic polymerization of trioxane [31]. A hybrid structure of FCC and ECC also can be found from the molten POM on the needle-shaped POM single crystal [32]. Many groups found obvious bands characteristic of ECC and FCC in the region of 1200–700 cm$^{-1}$ [33]. FTIR spectra were collected during the whole ageing process, as shown in figure 5 which shows the two bands only assigned to ECC and FCC [19]. The intensity of the bands characteristic of ECC and FCC both increase slightly with time, indicating that both the crystallinity of ECC and FCC increase during the ageing from 0 to 3 days. The crystallinity turns out to be higher, as shown in figure 4a, because of the increased crystallinity of ECC and FCC and the perfection of crystal structure [34]. We also can observe in figure 5 that the band of ECC increased faster than FCC, in agreement with the previous report [34].

In region II, the intensity of the bands for ECC and FCC increases slightly with time, indicating that both ECC and FCC go on increasing during the ageing from 3 to 10 days and ECC still grows faster than FCC. Both the ECC and FCC were unchanged in region III. These results are in good agreement with the WAXD tests.

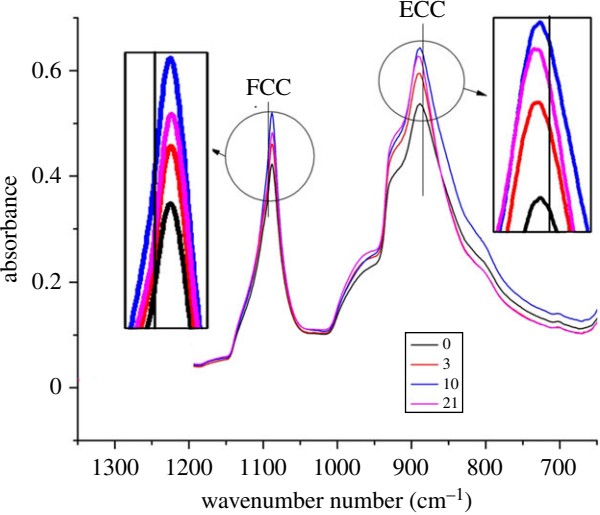

**Figure 5.** IR spectrum of POM aged 0 to 21 days in the region 1350–650 cm$^{-1}$.

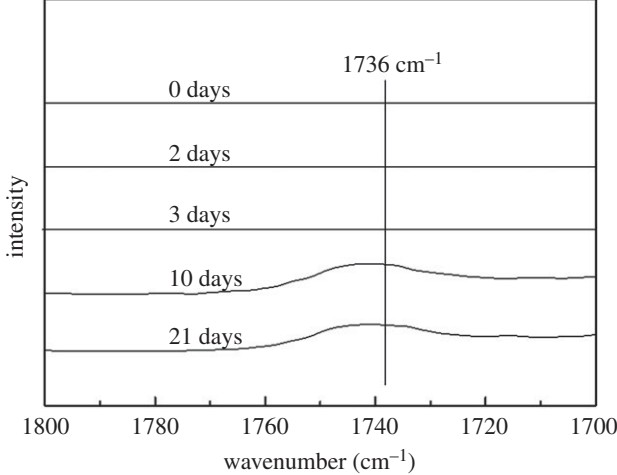

**Figure 6.** The change of carbonyl group that aged for a different time in the region 1800–1700 cm$^{-1}$.

### 3.3.2. Carbonyl group

In FTIR spectra, the characteristic peak of carbonyl group appears in the region of 1800–1700 cm$^{-1}$. For POM, as the product of thermal-oxidative degradation [33], it appears at 1736 cm$^{-1}$. However, in region I, as shown in figure 6, there is not any absorption peak, showing that chain scission does not happen during this period.

The ageing from 3 to 10 days is the region II. In figure 6, an absorption peak of carbonyl group appears. Although it is low, this change indicates that some of the molecular chains of POM copolymer are broken during the region II. As has been reported, chain scission takes place in the amorphous region in the ageing of POM and some of the broken chains form ECC or FCC, leading to an increase of crystallinity [34]. In other words, chemi-crystallization occurred at the amorphous region during the ageing from 3 to 10 days. In the region III, no further chain scission happened.

### 3.4. DSC analysis

Figure 7 gives the melting points ($T_m$) of the samples during ageing. As reported, the $T_m$ increases with the increase of grain size [35]. In region I, the $T_m$ increases, then decreases in region II, finally staying stable in region III. As displayed in figure 7, the change trend of $T_m$ stays the same with the variation of grain size (shown in figure 4b) exactly.

It is well known that the adding of stabilizers can slow down the thermal ageing of POM [6], the onset melting temperature ($T_0$) is a common method to judge the beginning of stabilizers

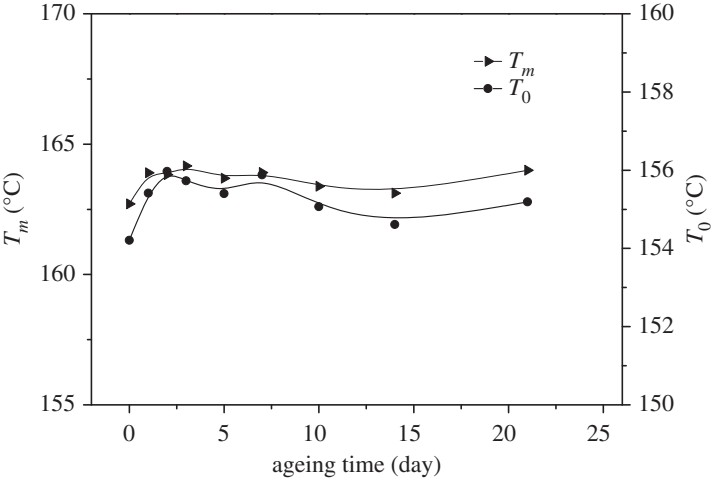

**Figure 7.** The change of melting point ($T_m$) and onset melting temperature ($T_0$) by DSC measurement.

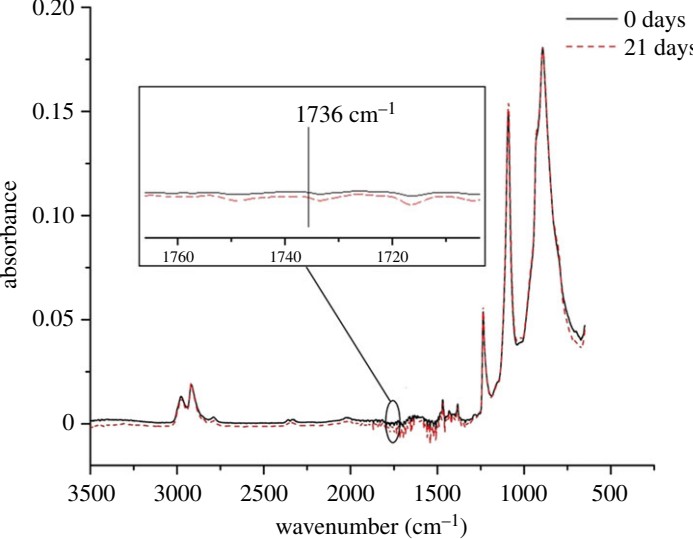

**Figure 8.** The change of core layer of aged 0 and 21 days.

consumption [3]. As shown in figure 7, the change of the onset melting temperature indicates that there is no stabilizer consumption in the whole ageing process, because if there is stabilizer consumption, the onset melting temperature will not rise again [7,36]. Archodoulaki *et al.* also pointed out that there was no stabilizers consumption even in the ageing under 100°C for 70 days [7].

## 3.5. The difference between skin and core

In injection-moulded articles, the specimen can generally be divided into three layers, the amorphous layer, the alignment layer and the spherulite layer. The thickness of amorphous layer will be lowered with increasing mould temperature. When the temperature reached 80°C, the thickness of amorphous can reach approximately 20 µm [37]. In this work, we also invested structural changes in the core layer and the skin layer of the specimens.

Figure 8 is the FTIR spectra of the core layer. It can be seen that there is no change in the whole graph and no carbonyl peak appears. Combining with the previous results, we can conclude that the thermal-oxidative degradation of POM copolymer only occurs in the amorphous skin layer.

As core layer has no shift, we just invest the difference between of core and skin that unaged, it can represent the change of core and skin for the whole ageing process.

The assignments of the FTIR bands of POM are listed in table 1 [31]. Figure 9 is the FTIR spectra of the skin and core layer for the unaged POM. We can see that there is no shift of the absorption peaks but changes in the peak height, which was related to the concentration of the groups in the system. It can be noticed that the difference in the intensity of the absorption peaks for the skin layer and the core

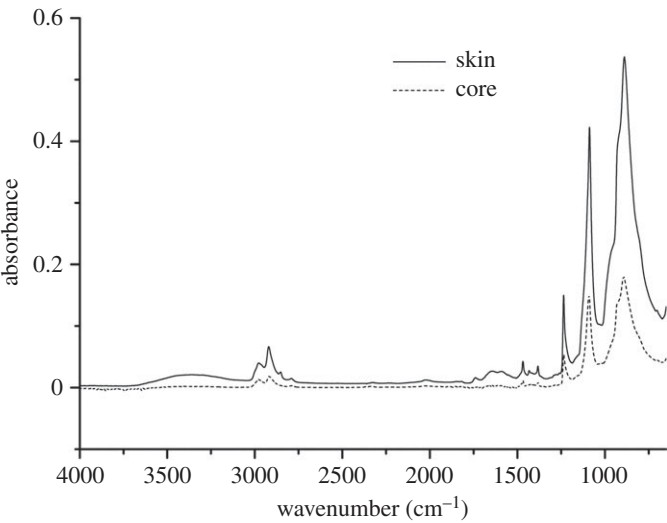

**Figure 9.** Skin and core layer for unaged POM.

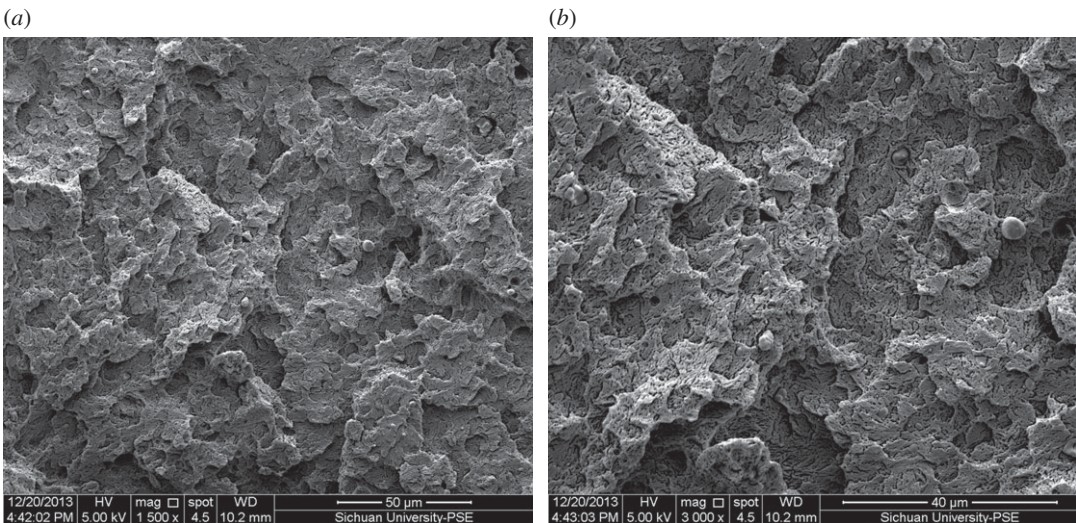

**Figure 10.** SEM of POM for skin (*a*) and core (*b*), ageing for 21 days.

**Table 1.** The static IR spectra bands assignments of POM.

| wavenumber (cm$^{-1}$) | assignments |
|---|---|
| 2945 | CH$_2$ asymmetric stretching |
| 1237 | CH$_2$ rocking + C-O-C bending + C-O-C symmetric stretching |
| 1134 | C-O-C antisymmetric stretching + O-C-O bending |
| 1093 | C-O-C stretching |
| 933 | C-O-C symmetric stretching |
| 903 | C-O-C antisymmetrics stretching + CH$_2$ rock |
| 633 | O-C-O bending, only belong to trigonal crystal |

layer is significantly larger than the difference in the intensity of the absorption peaks for the aged and unaged samples (figure 5). Also, it can be seen that the chain movement (including C-O-C, O-C-O, CH$_2$) in the skin layer was much fiercer than that in the core layer, due to the existence of residual stress. Second, the big difference indicated that skin layer has much more deficiency than the core layer.

According to figures 8, 9 and 10, we can conclude that the thermal-oxidative degradation of POM copolymer only occurs in the amorphous skin layer.

The whole oxidative ageing processes of POM are illustrated in figure 11. The crystalline region is made up of ECC and FCC with the little amorphous region and mainly exists in the core layer, while

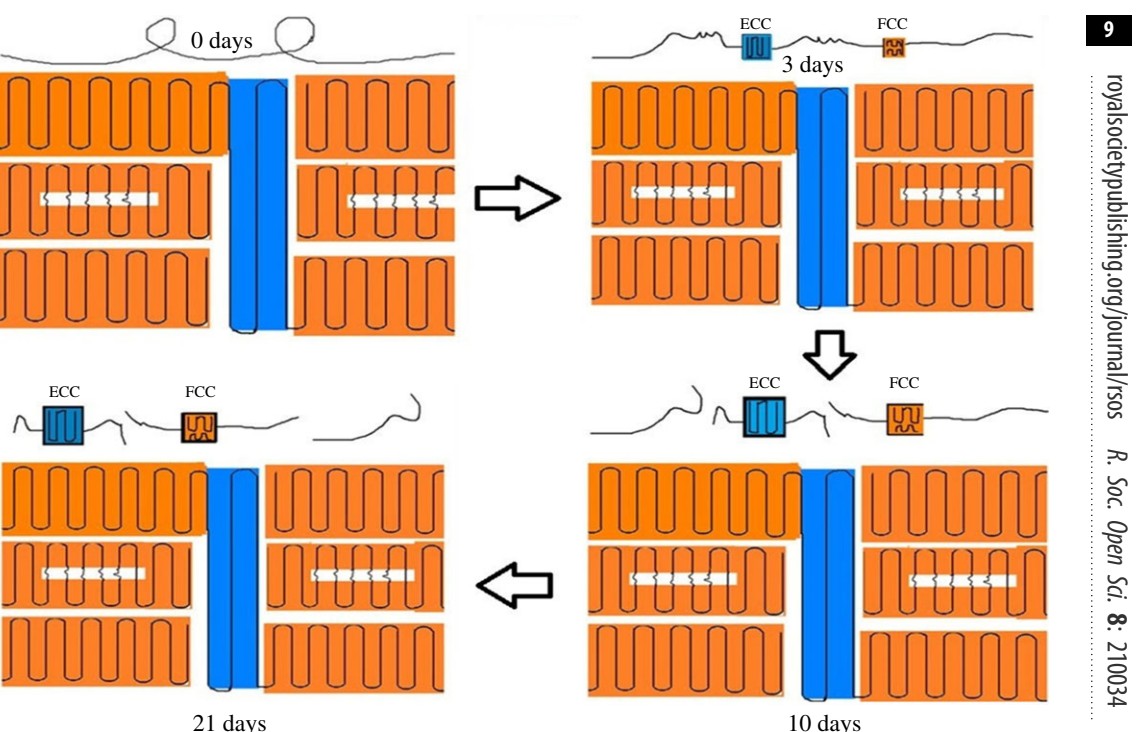

**Figure 11.** Whole thermal-oxidative ageing processes of POM. The cerulean areas represent ECC, and the yellow areas are FCC.

the amorphous region was mainly in the skin layer. After ageing for 3 days, the rearrangement of chains in the amorphous region into the crystal structure leads to the formation of FCC and ECC in the skin layer, and ECC has a higher growth rate than FCC. After ageing for 10 days, chain scission leads to chemi-crystallization, the content of both ECC and FCC becomes higher, and ECC also has a higher growth rate. In the region III, the structure has reached a stable state.

## 4. Conclusion

WAXD, FTIR, DSC and tensile test were employed to study the early thermal-oxidative ageing process of POM. According to the change of mechanical properties, the early thermal-oxidative ageing of POM copolymer was divided into three regions. The first region is ageing for 3 days and this is a heat treatment process. In this process, molecular chain arranges again but scission has not happened, both ECC and FCC increases in this region, and ECC grows faster than FCC. The second region is aged 3 to 10 days. It turns out to be chemi-crystallization process. The region III is aged 10 to 21 days, POM reached a stable situation. There were no stabilizers consumption, and no embrittlement happened in the whole process.

At last, we can be sure that thermal-oxidative degradation only occurred in the skin of amorphous layer, chain movement of skin is much more fierce than core, and skin has much more deficiency than core.

Data accessibility. The raw data are uploaded to the Dryad Digital Repository: https://datadryad.org/stash/share/sf2eTNNZp2asse0EFGhpdZWJ29cxt0CT4M-4CmbV2xA [38].

Authors' contributions. J.C. conceived this work; Y.-J.P. designed the work, analysed data and wrote the manuscript. W.-S.X., B.-T.Y. and H.-Y.N. assisted in designing the work, analysing data and revised the manuscript. All authors read and approved the final manuscript to be published.

Competing interests. We declare we have no competing interests.

Funding. We declare our manuscript has no funding supported.

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
