## [Peer Review File · Royal Society Open Science]

Review History

RSOS-210034.R0 (Original submission)

Review form: Reviewer 1

Is the manuscript scientifically sound in its present form?

No

Are the interpretations and conclusions justified by the results?

Yes

Is the language acceptable?

Yes

Do you have any ethical concerns with this paper?

No

Have you any concerns about statistical analyses in this paper?

No

Recommendation?

Accept with minor revision (please list in comments)

Comments to the Author(s)

No Figure of XRD can be seen, please add.

Review form: Reviewer 2**Is the manuscript scientifically sound in its present form?**

Yes

Are the interpretations and conclusions justified by the results?

Yes

Is the language acceptable?

Yes

Do you have any ethical concerns with this paper?

No

Have you any concerns about statistical analyses in this paper?

No

Recommendation?

Major revision is needed (please make suggestions in comments)

Comments to the Author(s)

This manuscript aims to investigate influence of early thermal-oxidative ageing on the structure and properties of polyoxymethylene copolymer. Obtained results showed that the early thermal-oxidative ageing of POM copolymer can be divided into three regions. The difference between the core layer and skin layer was also considered. The major problem is that the authors should add more analytical characterization experiments throughout the manuscript before it can be considered for reception.

1. Please revise language of your article by the native English.
2. In the introduction, there is a large amount of content about the influence of thermal aging on POM, but there is a lack of introduction of the influence of thermal aging on POM copolymer. Commercial POM copolymer containing 5% polytetrafluoroethylene powder and silicone oil. Do PTFE and silicone oil affect POM thermal aging?
3. To highlight the importance of this manuscript, the recent progress on the effect of early thermal-oxidative ageing of POM on performance should be added in the introduction.
4. During the ageing of POM at the temperatures below its melting points (usually between 80°C and 140 °C), properties generally show a relatively significant change before 500 hours. The authors only select the 100 °C as experiment aging temperature, the origin of the choice of aging temperature (100°C) needs to be clarified more clearly. Do the structure and properties change the same at other temperatures?
5. In the 2.3 characterization about mechanical properties, it is not enough to understand the change in properties of the sample by the tensile strength and elongation at break, more mechanical performance tests should be added.
6. Mechanical properties depend on chemical structure, molecular weight and distribution, branched and cross-linked, stress concentration, crystallization and orientation, and so on. The authors stated that the first three factors were not changed or happened in this period obviously and stress concentration was released, but relevant scientific evidence is insufficient.

7. In Fig.5, the authors stated that there is not any absorption peak in region I, showing that chain scission does not happen during this period. It is not enough to understand the change in structure of the sample by FTIR. Therefore, more characterizations should be added.

8. In Fig.6, the change of melting point (T_m) and onset melting temperature (T_0) is very small, should DSC test accuracy be considered?

9. In 3.5 the authors only according to the FTIR spectra of the skin and core layer, conclude that the thermal oxidative degradation of POM copolymer only occurs in the amorphous skin layer. Similarly, it is not enough to understand the change in structure of the sample by FTIR, more characterizations should be added. At least, Raman, and SEM must be measured to show the results.

Decision letter (RSOS-210034.R0)

Dear Dr Chen:

Title: Influence of early thermal-oxidative ageing on the structure and properties of polyoxymethylene copolymer
Manuscript ID: RSOS-210034

The editor assigned to your manuscript has now received comments from reviewers. We would like you to revise your paper in accordance with the referee and Subject Editor suggestions which can be found below (not including confidential reports to the Editor). Please note this decision does not guarantee eventual acceptance.

Please submit your revised paper before 24-Mar-2021. Please note that the revision deadline will expire at 00.00am on this date. If we do not hear from you within this time then it will be assumed that the paper has been withdrawn. In exceptional circumstances, extensions may be possible if agreed with the Editorial Office in advance. We do not allow multiple rounds of revision so we urge you to make every effort to fully address all of the comments at this stage. If deemed necessary by the Editors, your manuscript will be sent back to one or more of the original reviewers for assessment. If the original reviewers are not available we may invite new reviewers.

On behalf of the Subject Editor Professor Anthony Stace and the Associate Editor Professor Chaohua Cui.

RSC Associate Editor:
Comments to the Author:
(There are no comments.)

RSC Subject Editor:
Comments to the Author:
(There are no comments.)

Reviewers' Comments to Author:
Reviewer: 1

Comments to the Author(s)
No Figure of XRD can be seen, please add.

Reviewer: 2

Comments to the Author(s)

This manuscript aims to investigate influence of early thermal-oxidative ageing on the structure and properties of polyoxymethylene copolymer. Obtained results showed that the early thermal-oxidative ageing of POM copolymer can be divided into three regions. The difference between the core layer and skin layer was also considered. The major problem is that the authors should add more analytical characterization experiments throughout the manuscript before it can be considered for reception.

1. Please revise language of your article by the native English.
2. In the introduction, there is a large amount of content about the influence of thermal aging on POM, but there is a lack of introduction of the influence of thermal aging on POM copolymer. Commercial POM copolymer containing 5% polytetrafluoroethylene powder and silicone oil. Do PTFE and silicone oil affect POM thermal aging?
3. To highlight the importance of this manuscript, the recent progress on the effect of early thermal-oxidative ageing of POM on performance should be added in the introduction.
4. During the ageing of POM at the temperatures below its melting points (usually between 80°C and 140 °C), properties generally show a relatively significant change before 500 hours. The

authors only select the 100 °C as experiment aging temperature, the origin of the choice of aging temperature (100°C) needs to be clarified more clearly. Do the structure and properties change the same at other temperatures?

5. In the 2.3 characterization about mechanical properties, it is not enough to understand the change in properties of the sample by the tensile strength and elongation at break, more mechanical performance tests should be added.

6. Mechanical properties depend on chemical structure, molecular weight and distribution, branched and cross-linked, stress concentration, crystallization and orientation, and so on. The authors stated that the first three factors were not changed or happened in this period obviously and stress concentration was released, but relevant scientific evidence is insufficient.

7. In Fig.5, the authors stated that there is not any absorption peak in region I, showing that chain scission does not happen during this period. It is not enough to understand the change in structure of the sample by FTIR. Therefore, more characterizations should be added.

8. In Fig.6, the change of melting point (T_m) and onset melting temperature (T_0) is very small, should DSC test accuracy be considered?

9. In 3.5 the authors only according to the FTIR spectra of the skin and core layer, conclude that the thermal oxidative degradation of POM copolymer only occurs in the amorphous skin layer. Similarly, it is not enough to understand the change in structure of the sample by FTIR, more characterizations should be added. At least, Raman, and SEM must be measured to show the results.

Author's Response to Decision Letter for (RSOS-210034.R0)

See Appendix A.

RSOS-210034.R1 (Revision)

Review form: Reviewer 1

Is the manuscript scientifically sound in its present form?

Yes

Are the interpretations and conclusions justified by the results?

Yes

Is the language acceptable?

Yes

Do you have any ethical concerns with this paper?

Yes

Have you any concerns about statistical analyses in this paper?

Yes

Recommendation?

Accept as is

Comments to the Author(s)

No

Review form: Reviewer 2**Is the manuscript scientifically sound in its present form?**

Yes

Are the interpretations and conclusions justified by the results?

Yes

Is the language acceptable?

Yes

Do you have any ethical concerns with this paper?

No

Have you any concerns about statistical analyses in this paper?

No

Recommendation?

Accept as is

Comments to the Author(s)

I have no question.

Decision letter (RSOS-210034.R1)

Dear Dr Chen:

Title: Influence of early thermal-oxidative ageing on the structure and properties of polyoxymethylene copolymer
Manuscript ID: RSOS-210034.R1

It is a pleasure to accept your manuscript in its current form for publication in Royal Society Open Science. The chemistry content of Royal Society Open Science is published in collaboration with the Royal Society of Chemistry.

Please see the Royal Society Publishing guidance on how you may share your accepted author manuscript at <https://royalsociety.org/journals/ethics-policies/media-embargo/>. After publication, some additional ways to effectively promote your article can also be found here

<https://royalsociety.org/blog/2020/07/promoting-your-latest-paper-and-tracking-your-results/>.

On behalf of the Subject Editor Professor Anthony Stace and the Associate Editor Professor Chaohua Cui.

RSC Associate Editor:
Comments to the Author:
(There are no comments.)

RSC Associate Editor:
Comments to the Author:
(There are no comments.)

Reviewer(s)' Comments to Author:
Reviewer: 2

Comments to the Author(s)
I have no question.

Reviewer: 1

Comments to the Author(s)
No

Appendix A

Response to Reviewers

Dear Editor,

We are very thankful to the reviewers for good suggestions. According to the suggestions, the manuscript has been carefully revised, and the corrections made in the text have been marked in red. The responses to reviewers are as follows.

Response to Reviewer 1

1. No Figure of XRD can be seen, please add.

The Figure of XRD can be seen in Fig.3.

Fig.3. WAXD diffraction patterns of POM copolymer after ageing for various time.

Response to Reviewer 3

1. Please revise language of your article by the native English.

The author pays great attention to the revision of the paper to be sure to use native English

2. In the introduction, there is a large amount of content about the influence of thermal aging on POM, but there is a lack of introduction of the influence of thermal aging on POM copolymer. Commercial POM copolymer containing 5% polytetrafluoroethylene powder and silicone oil. Do PTFE and silicone oil affect POM thermal aging?

Due to the wide application and research of Co-POM, everyone agrees the methods of

improving the stability, early thermal oxidative and long-term ageing of POM are summarized and described from [4-9].

The main functions of 5% PTFE and silicone oil are reinforcement and lubrication, they are very common in POM additives and have little effect on thermal oxidative aging. The M90-44 are often used as standard products for testing.

3. To highlight the importance of this manuscript, the recent progress on the effect of early thermal-oxidative ageing of POM on performance should be added in the introduction.

There is no latest research progress on the early thermal oxidative aging of POM. The current research point is mainly to improve the aging conditions, such as adding hydrogen peroxide solution and performing ultraviolet irradiation to accelerate the performance change of polyoxymethylene. Therefore, it is very necessary for everyone to understand the effect of early thermal oxidative aging on the structural changes of POM

4. During the ageing of POM at the temperatures below its melting points (usually between 80°C and 140 °C), properties generally show a relatively significant change before 500 hours. The authors only select the 100 °C as experiment aging temperature, the origin of the choice of aging temperature (100°C) needs to be clarified more clearly. Do the structure and properties change the same at other temperatures?

The choice of ageing temperature is mainly considered from the perspective of use. The heat distortion temperature of POM is 110 °C , so the set temperature should be less than 110 °C . Secondly, the long-term use temperature of POM is -40°C-100°C (according to the manufacturer's instructions). The structure and performance will also change at other temperatures.

In order to obtain experimental data relatively quickly, we set 100°C as the ageing temperature.

5. In the 2.3 characterization about mechanical properties, it is not enough to understand the change in properties of the sample by the tensile strength and elongation at break, more

mechanical performance tests should be added.

It has been added in Fig.2.

6. Mechanical properties depend on chemical structure, molecular weight and distribution, branched and cross-linked, stress concentration, crystallization and orientation, and so on. The authors stated that the first three factors were not changed or happened in this period obviously and stress concentration was released, but relevant scientific evidence is insufficient.

In the region I, combined with the increase in grain size, makes the crystallization more perfect, coupled with the impact strength and tensile strength, reaching a high point, which is very consistent with the phenomenon of heat treatment. For heat treatment, it will not lead to changes in the first three factors.

7. In Fig.5, the authors stated that there is not any absorption peak in region I, showing that chain scission does not happen during this period. It is not enough to understand the change in structure of the sample by FTIR. Therefore, more characterizations should be added.

In the region I, no absorption peak is one of the phenomena. The tensile strength is in the process of increasing, which shows that the molecular chain will not break (if it breaks, it will cause the tensile strength to decrease). Also, the grain size is increase, we can confirm that the molecular chain is not broken

8. The lack of infrared absorption peak is. In Fig.6, the change of melting point (T_m) and onset melting temperature (T_0) is very small, should DSC test accuracy be considered?

We carefully confirmed the original data, and indeed got the data in the table, and the data is in line with the expected trend.

9. In 3.5 the authors only according to the FTIR spectra of the skin and core layer, conclude that the thermal oxidative degradation of POM copolymer only occurs in the amorphous skin layer. Similarly, it is not enough to understand the change in structure of the sample by FTIR, more characterizations should be added. At least, Raman, and SEM must be measured to show the results.

The SEM data has been added. Obviously, there is no change in the surface layer and the core layer. Combined with the FTIR data, it can be proved that it only occurs in the amorphous skin layer.